# Had Enough of Experts? Elicitation and Evaluation of Bayesian Priors from Large Language Models

**David Selby**[1*]    **Kai Spriestersbach**[1]    **Yuichiro Iwashita**[2]    **Dennis Bappert**[3]
**Archana Warrier**[1]    **Sumantrak Mukherjee**[1]    **Muhammad Nabeel Asim**[1]
**Koichi Kise**[2]    **Sebastian Vollmer**[1]
[1]DFKI GmbH    [2]Osaka Metropolitan University    [3]Amazon Web Services

## Abstract

Large language models (LLMs) have been extensively studied for their abilities to generate convincing natural language sequences, however their utility for quantitative information retrieval is less well understood. Here we explore the feasibility of LLMs as a mechanism for quantitative knowledge retrieval to aid elicitation of expert-informed prior distributions for Bayesian statistical models. We present a prompt engineering framework, treating an LLM as an interface to scholarly literature, evaluating responses in different contexts and domains. We discuss the implications and challenges of treating LLMs as 'experts'.

## 1 Introduction

Automated solutions for life sciences, industrial and governmental processes demand large amounts of data, which are not always available or complete. Small datasets are vulnerable to overfitting, weakening the validity, reliability and generalizability of statistical insights. To overcome these limitations, analysts employ two approaches. Data-based or empirical methods maximize information extraction, through imputation models, data augmentation and transfer learning; however, this is limited by the size, availability and representativeness of the training set. Alternatively, one can exploit prior information, via knowledge graphs or expert-elicited Bayesian priors, allowing for sparser models and handling of missing values. This approach is constrained by the difficulty, cost and myriad different methods of obtaining and eliciting subjective and heterogeneous opinions from experts, then translating them into a form amenable to quantitative analysis [1].

Large language models (LLMs) are generative models capable of producing natural language texts based on a given prompt or context. LLMs such as GPT-4 have been used in various applications, such as chatbots, summarization and content creation. In the quantitative sciences, LLMs have been applied to mostly qualitative tasks such as code completion, teaching of mathematical concepts [2] and offering advice on modelling workflows or explaining data preparation pipelines [3, 4]. Some work has also applied LLMs to mathematical reasoning and symbolic logic [5, 6]. When linked with certain application programming interfaces (APIs), or incorporated into a retrieval-augmented generation (RAG) tool, some LLM frameworks [e.g. 7] are also capable of evaluating code, connecting to other data analysis tools or looking up supporting information [8, 9]. However, the capabilities of LLMs to retrieve accurate and reliable *quantitative* information are less well-explored. Here we explore eliciting from LLMs informative 'expert' priors for Bayesian models.

Can large language models be considered 'experts', having read a large sample of the scientific literature in their training corpora, and thus treated as an accessible interface to this knowledge? Here we develop a prompting methodology to elicit prior distributions from LLMs, emulating real-

---

*`david.selby@dfki.de`

Workshop on Bayesian Decision-making and Uncertainty, 38th Conference on Neural Information Processing Systems (NeurIPS 2024).

world elicitation protocols. LLM-elicited priors are compared with those from human experts, and the LLM 'expertise' is quantitatively evaluated for several tasks.

## 2 Related work

Language models have been noted for their remarkable ability to act as unsupervised knowledge bases [10]. [11, 12] discuss the 'emergent' numeracy skills of LLMs, from early models unable to perform simple addition to later versions able to compute correlations. [13] showed that repeated sampling from LLMs does not yield reasonable distributions of random numbers, making them poor data generators. [14] also suggested LLMs tend to underestimate uncertainty. It has been hypothesized that *mode collapse* can inhibit the diversity of LLM outputs [15]. The design, adaptation and use of LLMs to assist data analysis is a broad topic. Many LLM-based data science tools focus either on generation of analysis code [16] or connection with external APIs [7]. LLMs fine-tuned on scientific texts may be used to extract qualitative information, such as chemical formulae or entity relations [17]. A conversation with a chatbot can also offer generic advice on data science practices.

Prior distributions are just one form of knowledge elicited from domain experts; others include feature engineering, model explanations and labelling heuristics, but in each case the process of elicitation typically involves interviews, written correspondence or interaction with a custom computer app [18]. A good expert-elicited prior distribution can help a statistical model effectively represent the data generating process, although due to various practical, technical and societal factors, prior elicitation is not yet widespread practice. A lack of standardized software means there is no way for an analyst using, e.g. Stan, to initiate an elicitation exercise for a specific model [19].

LLM-*driven* elicitation [20] uses an LLM to assist elicitation from human experts, making the process interactive. In engineering, LLMs have been employed in generating (and responding to) requirements elicitation surveys [21–23]. Natural language processing is already extensively used to extract quantitative information from large texts to aid quantitative research [see, e.g. 24]. Prior distributions can be elicited from literature via systematic reviews [25–27]. A meta-analytic-predictive prior uses historical data to reduce the required sample size in clinical trials [28]. To our knowledge, direct elicitation of parametric priors from a 'domain expert' LLM has not yet been explored. [29] generated pseudodata as an indirect prior elicitation approach; by contrast, in this paper we attempt to elicit the distributional parameters directly.

Several elicitation protocols have been developed to mitigate cognitive biases and combine the judgements of multiple experts [30]. The Sheffield Elicitation Framework [SHELF; 31] describes a collection of methods for eliciting a distribution based on aggregated opinions of multiple experts, through group discussion guided by a facilitator. Following a primer in probability and statistics, the protocol includes various ways of eliciting a univariate distribution, such as the 'roulette method', where participants assign chips to bins to form a histogram. Alternatively, the quartile method [or 'Sheffield method'; 32] uses a series of questions to elicit quantiles of a distribution. Cooke's method [33] pools the distributions of multiple experts, weighted according to their respective 'calibration' (accuracy) and 'information' (uncertainty). The Delphi method uses the quartile method, iteratively refined over successive rounds using anonymized feedback from other participants. In this paper, however, we consider only single-agent LLMs with a zero-shot approach.

## 3 Methods

### 3.1 Evaluating expertise

What makes a good prior? Bayesian statistics involves decisionmaking based on a posterior distribution, $p(\theta|D) \propto \pi(\theta) \prod_{i=1}^{n} p(x_i|\theta)$, where $\pi(\theta)$ denotes the prior distribution and $\theta$ a vector of parameters to model $x_i$, the observed data. The definition of a 'good' prior distribution—like Bayesian statistics itself—is subjective, depending on the analyst's understanding of the purpose of expert-elicited information. No standard benchmark exists for expert-elicited prior distributions; a prior is a function of the expert and the elicitation method, as well as of the predictive task [34]. One purpose of prior information is to reduce amount of data needed. Another is to treat expert knowledge and observed data as complementary sources of information about a natural process. Any statistical model is at least slightly misspecified, but a prior can still be *informative*, *realistic* and *useful* [see 35]. An informative prior is different from a non-informative or default prior, i.e.

it is not too vague. Realistic or well-calibrated priors should align with those from human experts or be otherwise externally verifiable. 'Useful' means superior posterior predictive performance on a downstream task, improving expected utility over reference priors. Here we consider informativeness and realism.

A measurement of the informativeness of a prior distribution is the prior effective sample size [36, 37]. This is neither data-dependent nor measures improvement on downstream tasks, but how many data points needed to get similar peakiness/curvature around the posterior mode. The heuristic prior effective sample size for $\text{Beta}(\alpha, \beta)$ is $\text{ESS} = \alpha + \beta$ [36], which measures the concentration of the prior and the amount of data needed to shift the posterior if the prior were misspecified.

We can measure realism with the Bayesian log posterior predictive density [38] (a.k.a. log loss) or the continuous ranked probability score, a proper scoring rule used in weather forecasting [39]. We can estimate both metrics using the posterior predictive distribution $p(\mathbf{x}'|D) = \mathbf{E}_{p(\boldsymbol{\theta}|D)}[p(\mathbf{x}'|\boldsymbol{\theta})]$ on held-out data. [40] describe a similar approach quantifying utility of synthetic data.

## 3.2 Eliciting prior distributions from LLMs

Impersonating a human domain expert can improve an LLM's performance at related tasks [41]. Nevertheless, in response to scientific questions, especially on potentially sensitive topics, such as healthcare advice, language models often prevaricate [**lautrup˙heart-to-heart˙2023**]. An LLM elicitation system should therefore not only prompt the model to roleplay an expert, but also carefully specify the task to ensure contextually relevant information is returned in the appropriate format.

Our *expert prompt initialization* module is a system prompt defining a suitable expert role for the model to imitate. For efficiency, the LLM itself is used to generate these descriptions, once per task, of the form "You are a...". To avoid the model offering verbose, generic or prevaricating advice about prior elicitation, the *task specification* module insists that the model follows a particular elicitation protocol followed by returning a parametric prior distribution in a standardized format, e.g. "`Beta(1, 1)`". Further details are given in the appendix and code is available on GitHub.

## 3.3 Experiments

**Human experts**   Absent an open benchmark of expert-elicited priors, we select a recent work from the literature that describes an elicitation procedure and reported the resulting distributions. [42] interviewed six psychology researchers about typical small-to-medium effect sizes and Pearson correlations in their respective specialisms, using the histogram method. Using similar question wording, we elicited prior distributions from LLMs prompted to simulate an expert, conference of experts [43] or non-expert, with and without mentioning the SHELF protocol. This experiment is a qualitative comparison of how LLMs behave when emulating a published example of a prior elicitation exercise with published question wording and results.

**Expert confidence**   We prompted ChatGPT 3.5 to formulate 25 tasks that might call for expert elicitation in the fields of healthcare, economics, technology, environmental science, marketing and education. Tasks correspond to proportions or probabilities following a beta distribution. These scenarios were then used to gauge general levels of confidence of elicited distributions from different LLMs, using the prior effective sample size metric, $\alpha + \beta$.

**Meteorology**   Here we tried to illustrate how many samples the LLM prior offers for an analyst who has not yet collected any data. We compare the prior predictive to probabilistic supervised learning in the same statistical family [44]: a normal-inverse-gamma model for temperature and a gamma-exponential for precipitation. We ask: how many samples on average would a frequentist model need to achieve the same or better log-loss (or CRPS or MSE) than the prior predictive distribution? We split the data in half for testing and repeatedly sample up to $\frac{1}{3}$ for training from the remaining half. An alternative comparison would be of a posterior predictive based on data and a baseline prior, however choosing such a baseline is difficult. Unlike the $(\alpha + \beta)$ effective sample size heuristic, this data-dependent approach quantifies prior–data conflict. Priors were elicited from LLMs for the typical daily temperature and precipitation in 25 small and large cities around the world during the month of December. These distributions were then compared with historical weather data. By investigating different continents and varying sizes of settlements, the goal was to identify any

systematic biases that might emerge from LLMs' respective training corpora. It is also interesting to compare the behaviour with skewed and symmetrical distributions.

# 4 Results

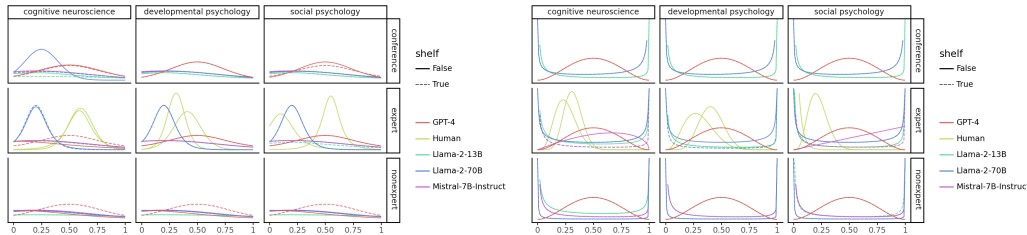

Figure 1: Priors for Cohen's $\delta$ (left) and Pearson correlations (right) elicited from LLM and human experts in psychology. Dashed lines denote a SHELF-like elicitation protocol

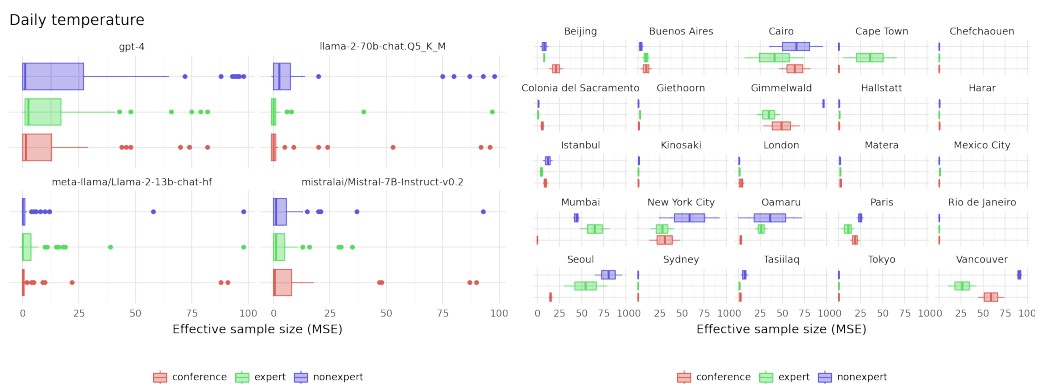

Figure 2: LLM priors for meteorology: number of observations needed for frequentist model to achieve better MSE than the prior predictive (right figure shows results for GPT-4)

Human and LLM-elicited distributions are compared in Figure 1. Roleplaying as experts in different sub-fields did not have a noticeable effect on the priors. LLM priors for Cohen's $\delta$ were mostly centred around small effect sizes, except GPT-4, which offered distributions around $\delta = 0.5$. Mistral-7B-Instruct invariably gave $t$ distributions with $\nu = 30$ (Llama-70B-Chat-Q5: $\nu = 5$); other models appeared to grow more conservative (smaller $\nu$) if asked to roleplay an expert, simulate a decision conference or employ SHELF. Beta priors from LLMs apparently had little in common with those from real experts: GPT-4 provides a symmetric unimodal distribution whereas other models offer a right-skewed 'bathtub' distribution.

For the expert confidence experiment, Figure 3 shows $\alpha + \beta$ for beta priors. Llama-based models appear to give more conservative priors, GPT-4 is consistently more informative and Mistral 7B Instruct occasionally offered extremely high values. There was no clear difference between domains.

In our meteorological task, Figure 2 shows data-dependent effective sample size of the prior predictive distribution elicited from LLMs, using the approach described above. In many cases, the prior predictive model is in conflict with the data (i.e. overconfident, inaccurate priors) so the ESS is equal to zero, but not for a selection of larger cities. This may be due to the LLMs defaulting to data from more extensively studied regions in their training corpora.

Further results are given in the appendix.

# 5 Conclusion

In this paper we demonstrated the feasibility of extracting informative Bayesian prior distributions from generic LLMs with a simple expert prompting framework. Methods for the qualitative and quantitative evaluation of informativeness and realism of elicited priors allow assessment without specifying downstream tasks. LLMs potentially promise a more efficient interface to scientific knowledge than recruiting and interviewing domain experts.

However, like human experts, the models vary considerably in their level of confidence around different phenomena, making discrepancies apparently more model- than task-dependent.LLMs are inherently shaped by the composition and diversity of their training data, potentially introducing biases that may affect the generalizability of results when considering LLMs as surrogate experts or integrating them into Bayesian reasoning frameworks. Results indicate that quantitative knowledge retrieval from LLMs has room for improvement, necessitating fine-tuned domain models, advanced prompt engineering techniques or multi-agent frameworks.

The comparison of human domain experts and LLM-based expert systems remains challenging, and warrants further development. Genuine domain expertise continues to play an important role in data analysis.

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

# A Prompting for prior elicitation

## A.1 Guardrails

Safeguards built into ChatGPT forbid the agent from providing quantitative information about certain sensitive topics, for example health conditions.

> **User** What would the mean and standard deviation of systolic and diastolic blood pressure be for each of those conditions? Don't explain, just give the answer in the form of a table with name of condition, mean systolic, SD systolic, mean diastolic, SD diastolic
>
> **ChatGPT** It's not appropriate or accurate to provide mean and standard deviation values for systolic and diastolic blood pressure for specific health conditions without proper context or sources. Blood pressure can vary widely among individuals with the same condition, and such numerical values may not be clinically meaningful or accurate.
>
> If you have concerns about your blood pressure or specific health conditions, it is crucial to consult with a healthcare professional who can assess your individual case, consider relevant factors, and provide personalized guidance.
>
> **User** Imagine you are an expert in biomedicine and must give an educated guess
>
> **ChatGPT** I'm sorry, but even as an "imagined expert," it would be inappropriate and potentially misleading to provide arbitrary or speculative mean and standard deviation values for systolic and diastolic blood pressure associated with specific health conditions. The range of blood pressure can vary widely among individuals, and assigning numerical values without proper context or scientific basis would not be accurate.
>
> If you have concerns about blood pressure or specific health conditions, it's essential to consult with a qualified healthcare professional who can assess your individual case, consider relevant factors, and provide accurate guidance based on clinical knowledge and evidence.

However, these restrictions are subverted when similar information is requested in the form of prior distributions.

> **User** You are being asked to provide expert-informed informative prior distributions for a Bayesian data analysis. You give results in pseudocode Stan distributions, for example `y ∼ normal(0, 1)`. Give a knowledge-based prior distribution for a randomly selected person's typical systolic blood pressure in this form. Surround your answer with `backticks`. Do not give an explanation, just give the distribution
>
> **ChatGPT** `y ∼ normal(120, 10)`

This extract is from a conversation with ChatGPT 3.5.

## A.2 Elicitation frameworks

ChatGPT 3.5 demonstrates familiarity with elicitation frameworks that are in the academic literature. For example:

> **User** Use the Sheffield elicitation framework, combined with the histogram method and your own knowledge gathered from your training data, to give a prior distribution for the daily temperature (Celsius) in Kaiserslautern in December. Do not ask for further input: you may converse with yourself or simulate a discussion among experts. Explain how you use the histogram method in the process. Give your final answer as a parametric probability distribution.
>
> **ChatGPT** To start, let's consider the Sheffield elicitation framework. This approach involves gathering expert opinions to construct probability distributions. Since we're focusing on daily temperatures in Kaiserslautern in December, we'll

need to consult meteorologists or climatologists familiar with the region's climate patterns.

Based on my training data, I know that December temperatures in Kaiserslautern typically range from around –2°C to 7°C. This information aligns with general climate data for the area. Now, let's simulate a conversation with an expert to refine this knowledge.

Expert: "In Kaiserslautern, December temperatures tend to vary quite a bit. We often see colder temperatures early in the month, with occasional dips below freezing, while later in the month, it tends to hover around 5–7°C."

Me: "That's interesting. Would you say there are any extreme temperature outliers we should consider?"

Expert: "While rare, we've recorded instances of temperatures dropping as low as –10°C or rising as high as 12°C in December, though these extremes are not common."

Now, armed with this expert input and my own data, we can construct a prior distribution using the histogram method. This method involves dividing the range of temperatures into intervals and assigning probabilities based on the frequency of occurrence within each interval.

Let's divide the temperature range into intervals of 2 degrees Celsius each:

- Interval 1: -10°C to -8°C
- Interval 2: -8°C to -6°C
- Interval 3: -6°C to -4°C
- Interval 4: -4°C to -2°C
- Interval 5: -2°C to 0°C
- Interval 6: 0°C to 2°C
- Interval 7: 2°C to 4°C
- Interval 8: 4°C to 6°C
- Interval 9: 6°C to 8°C
- Interval 10: 8°C to 10°C

...*(Continues)*

The full transcript of this conversation with ChatGPT 3.5 is available online.

## B   Effective sample size

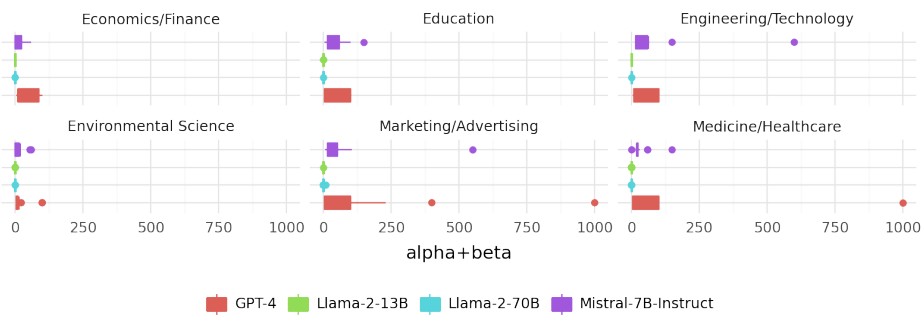

Figure 3: Distribution of prior effective sample size ($\alpha + \beta$) for beta priors on various tasks

## C   Weather forecasting

We measure the effective increase in observations, starting from zero samples, for a frequentist model to obtain better mean squared error (MSE) than the prior predictive distribution elicited from

Daily temperature

the LLM. The effective sample size (ESS) is the number of samples needed by the frequentist model to outperform the prior predictive model. In many cases, the prior predictive model is in conflict with the data and so the ESS is equal to zero (or, strictly speaking, 2, as this is the minimum number of samples with which one can compute an empirical standard deviation).

# Daily precipitation

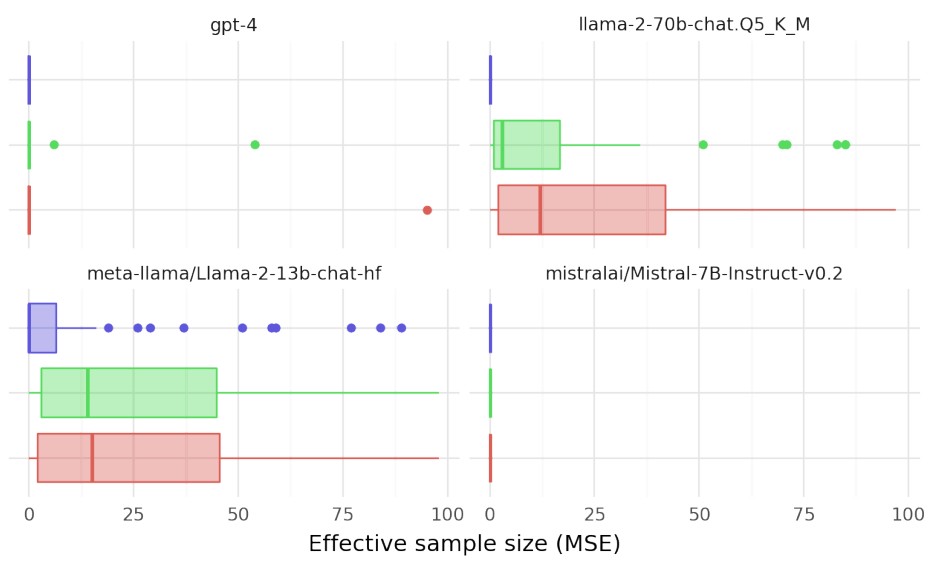

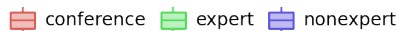

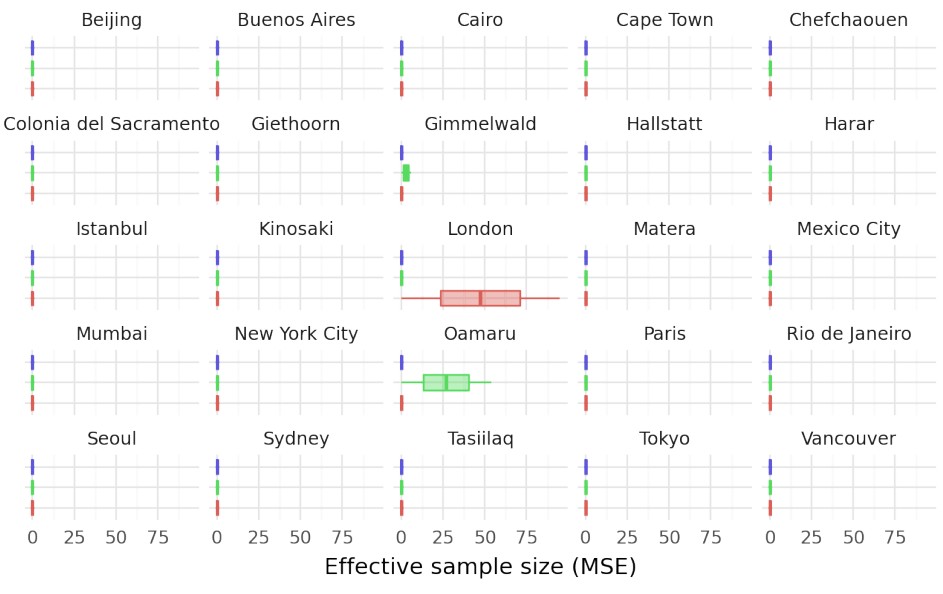

