# OpenReview forum: "Had enough of experts? Elicitation and evaluation of Bayesian priors from large language models"
_NeurIPS.cc/2024/Workshop/BDU — NeurIPS BDU Workshop 2024 Poster_

### Official Review · Reviewer_yL7e · 2024-09-24
**Method description and some evaluation details needs more clarity**

**Rating:** 4
**Confidence:** 3

**Review:**

This paper investigates the use of LLMs as experts to elicit prior distributions for Bayesian models.

The authors are looking at an interesting problem, given that:
- Prior elicitation is an important problem in many real-world problems with small datasets.
- LLMs are trained on so much information, one would be curious to know whether they have any utility in prior elicitation.

However, the exact prompt methodology is not very clear. In Section 1 (line 33), the authors mention they use a chain-of-thought based method, however in the Appendix (starting from line 370), where prompt examples are shown, there is no chain-of-thought prompting used.
In Section 3.2 (line 102), an 'expert prompt initialization' module and a 'task specification module' are mentioned, however the relation between these modules and the prompts shown in the Appendix A are very implicit.

In Figure 1, it is not clear what a 'conference of experts' is (Section 3.3 line 113), an explanation or citation would be appreciated.

For the Meteorology experiment, the 'probabilistic supervised learning model from the same statistical family' (line 123) is not described. Also, all the figures for this experiment have a noticeable typo "Effective Sample Size (MSE)". In both Section 4 (line 148) and the Appendix C (line 426), the authors mention the ESS is 0 (or technically 2) due to prior predictive being in conflict with the data. More detail on how this mismatch occurs would be good to know - this probably relates to describing the model used for this experiment as mentioned above. Also, it is not clear which of the 4 LLMs (GPT-4, Llama-2-70B, Llama-2-13B, Mistral-7B-Instruct) were used to get the 5x5 plots showing the ESS for each of the cities.

Some other issues include:
- Line 60 -> Claim that direct elicitation of priors from a domain expert LLM has not yet been explored, is incorrect. See e.g. https://openreview.net/forum?id=euLzlnU7gz. This is understandable given that the subfield of using prompts in LLMs for different tasks is moving fast.
- Line 82 -> A citation for the taxonomy of informativeness, realism and usefulness for priors would be appreciated.
- The term 'domain expert LLM' and 'expert LLM' seem to be used interchangeably. The reader may think that different prompts are used for them, so it would be good to be explicit on whether these refer to the same LLM agent.
- Line 99 -> Citations on prevaricating examples would be appreciated.

Some smaller issues which do not affect this evaluation of the work:
- Line 29, 31 -> large language models -> LLMs - this is for consistency.
- Line 69 -> Missing fullstop before "The Delphi...".
- Line 78 -> "No standard benchmark exists for expert-elicited prior distributions" -> This should refer to *evaluating* expert-elicited prior distributions, not expert-elicited prior distributions themselves.
- Line 82 -> misspecified, Line 91 -> mis-specified - this is for consistency.
- Line 145 -> GPT -> GPT4.
- Line 158 -> "... and a discrepancy apparently more model- than task-dependent" -> sentence could be written better.

---

### Official Review · Reviewer_BNqZ · 2024-10-01
**The manuscript innovatively explores the use of LLMs for Bayesian prior elicitation with a robust methodology, though it requires minor revisions for deeper discussion on biases, expanded analyses, and refinements for clarity and comprehensive literature coverage.**

**Rating:** 8
**Confidence:** 4

**Review:**

The manuscript titled "Had Enough of Experts? Elicitation and Evaluation of Bayesian Priors from Large Language Models" presents an intriguing exploration into the use of Large Language Models (LLMs) for eliciting expert-informed priors in Bayesian statistical models. The study is innovative in its approach to leveraging LLMs as a novel source of quantitative information, which is a significant departure from traditional expert elicitation methods. The research question addresses a gap in the field of statistical analysis and data science, where the integration of AI-generated priors could potentially streamline the process of model building. The methodology is well-constructed, with a clear framework for eliciting priors from LLMs and a systematic comparison with human experts. The use of a prompt engineering framework and the evaluation against actual expert-elicited priors are strengths of the study. However, the paper would benefit from a more detailed discussion on the potential biases introduced by the training data of the LLMs and how these might affect the generalizability of the results. The data analysis appears to be thorough, with appropriate statistical measures used to evaluate the informativeness and realism of the priors. The results are clearly presented, and the comparison with historical weather data is a clever way to test the LLMs' predictive capabilities. Nevertheless, the paper could be improved by including additional analyses that explore the impact of different model architectures or training datasets on the elicited priors.

Overall, the manuscript is a valuable contribution to the field, offering a fresh perspective on the application of LLMs in statistical analysis. I recommend it for acceptance with minor revisions, including the expansion of the discussion on biases, additional analyses to explore model variability, and a refinement of the manuscript to enhance clarity and conciseness.

---

### Decision · Program_Chairs · 2024-10-09

Accept (Poster)